

# Mitochondria dysfunction is one of the causes of diclofenac toxicity in the green alga *Chlamydomonas reinhardtii*

Darya Harshkova[1], Elżbieta Zielińska[1], Magdalena Narajczyk[2], Małgorzata Kapusta[2] and Anna Aksmann[1]

[1] Department of Plant Experimental Biology and Biotechnology, Faculty of Biology, University of Gdansk, Gdansk, Poland
[2] Bioimaging Laboratory, Faculty of Biology, University of Gdansk, Gdansk, Poland

Corresponding author
Anna Aksmann,
anna.aksmann@ug.edu.pl

## ABSTRACT

**Background:** Non-steroidal anti-inflammatory drugs (NSAIDs), such as diclofenac (DCF), form a significant group of environmental contaminants. When the toxic effects of DCF on plants are analyzed, authors often focus on photosynthesis, while mitochondrial respiration is usually overlooked. Therefore, an *in vivo* investigation of plant mitochondria functioning under DCF treatment is needed. In the present work, we decided to use the green alga *Chlamydomonas reinhardtii* as a model organism.

**Methods:** Synchronous cultures of *Chlamydomonas reinhardtii* strain CC-1690 were treated with DCF at a concentration of 135.5 mg $\times$ L$^{-1}$, corresponding to the toxicological value EC50/24. To assess the effects of short-term exposure to DCF on mitochondrial activity, oxygen consumption rate, mitochondrial membrane potential (MMP) and mitochondrial reactive oxygen species (mtROS) production were analyzed. To inhibit cytochrome *c* oxidase or alternative oxidase activity, potassium cyanide (KCN) or salicylhydroxamic acid (SHAM) were used, respectively. Moreover, the cell's structure organization was analyzed using confocal microscopy and transmission electron microscopy.

**Results:** The results indicate that short-term exposure to DCF leads to an increase in oxygen consumption rate, accompanied by low MMP and reduced mtROS production by the cells in the treated populations as compared to control ones. These observations suggest an uncoupling of oxidative phosphorylation due to the disruption of mitochondrial membranes, which is consistent with the malformations in mitochondrial structures observed in electron micrographs, such as elongation, irregular forms, and degraded cristae, potentially indicating mitochondrial swelling or hyper-fission. The assumption about non-specific DCF action is further supported by comparing mitochondrial parameters in DCF-treated cells to the same parameters in cells treated with selective respiratory inhibitors: no similarities were found between the experimental variants.

**Conclusions:** The results obtained in this work suggest that DCF strongly affects cells that experience mild metabolic or developmental disorders, not revealed under control conditions, while more vital cells are affected only slightly, as it was already indicated in literature. In the cells suffering from DCF treatment, the drug influence on mitochondria functioning in a non-specific way, destroying the structure of mitochondrial membranes. This primary effect probably led to the mitochondrial inner membrane permeability transition and the uncoupling of oxidative

phosphorylation. It can be assumed that mitochondrial dysfunction is an important factor in DCF phytotoxicity. Because studies of the effects of NSAIDs on the functioning of plant mitochondria are relatively scarce, the present work is an important contribution to the elucidation of the mechanism of NSAID toxicity toward non-target plant organisms.

# INTRODUCTION

Non-steroidal anti-inflammatory drugs (NSAIDs) have become a significant group of environmental contaminants in water bodies (*Mulkiewicz et al., 2021*; *Ortúzar et al., 2022*). Diclofenac (DCF) is one of the most commonly detected pharmaceuticals in water samples worldwide (*Ortúzar et al., 2022*). DCF and other NSAIDs were designed as drugs for humans and animals, but their biological effects on non-target organisms, including higher plants and algae, are attracting increasing attention (*He et al., 2017*; *Ortúzar et al., 2022*). Nevertheless, precise ecotoxicological information is insufficient.

One of the consequences of plant cells' exposure to anthropogenic contaminants, among them pharmaceuticals, is the disruption of cell metabolism and induction of oxidative stress (*Schmidt & Redshaw, 2015*; *He et al., 2017*; *Hejna, Kapuścińska & Aksmann, 2022*). When the toxic effects of DCF on plants are analyzed, authors often focus on the most distinctive biomarkers like photosynthetic efficiency parameters (*Copolovici et al., 2017*; *Majewska et al., 2018, 2021*; *Hájková et al., 2019*; *Svobodníková et al., 2020*), while another very important process, mitochondrial respiration, is usually overlooked. Bearing in mind that chloroplast-mitochondria interaction plays an important role in general cell metabolism, as well as in stress tolerance (*Van Aken et al., 2009*), investigation of mitochondria functioning under DCF treatment is of special interest. The problem is even more interesting because the respiratory electron transport chain functions analogously to the photosynthetic electron transport chain, and there is a lot of evidence that the latter is disrupted under DCF treatment (*Kummerová et al., 2016*; *Hájková et al., 2019*; *Ben Ouada et al., 2019*). Thus, it is likely that not only chloroplasts but also plant mitochondria are susceptible to this drug. This assumption is supported by the observations that DCF causes changes in the functioning of mitochondria in animal cells, such as inhibition of mitochondrial complex III, decrease of mitochondrial membrane potential, increase of the *Bax*, *cytochrome c*, *cas-3*, *cas-8* and *p53* expression at gene transcription level (*Ghosh et al., 2016*; *Darendelioglu, 2020*) and by our preliminary observations of changes in mitochondrial membrane potential (MMP) in DCF-treated cells of the green alga *Chlamydomonas reinhardtii* (*Harshkova, Zielińska & Aksmann, 2019*). The value of MMP, generated by the proton pumps (complexes I, III and IV) may provide some clues to the mitochondria's ability to produce ATP (*Zorova et al., 2018*). MMP plays an important role also in mitochondrial ion transport (*Zorova et al., 2018*) and

in the regulation of metabolic processes. The MMP value is relatively stable under homeostasis, its change (decrease or increase) could serve as a valuable indicator of substance toxicity and the physiological response to stress in unicellular plant organisms.

The mitochondrial inner membrane of green algae and plants contains a standard oxidative phosphorylation system with electron transport chain (ETC) complexes (complexes I–IV) and ATP synthase (often named complex V) (*Møller, Rasmusson & Van Aken, 2021*). However, the existence of an alternative route of electron transport in plant mitochondria, especially the presence of cyanide-resistant alternative oxidase (AOX) (*Møller, Rasmusson & Van Aken, 2021*), makes investigations of plant mitochondria relatively complicated. On the one hand, the presence of AOX forms a branch in the ETC, partitioning electrons between the cytochrome *c* pathway and the AOX pathway. The consequence of high AOX activity is a significant modulation of MMP and a decrease in mitochondrial energy production (ATP yield) (*Vanlerberghe, 2013*). On the other hand, the channeling of electrons into AOX prevents ROS overproduction under stress conditions (*Vanlerberghe, Cvetkovska & Wang, 2009*), thus, electron distribution between the cytochrome *c* pathway and the AOX pathway can be regarded as one of the most important factors in plant response to environmental pollutants, including NSAIDs. Literature reports suggest that DCF in concentrations up to 12 mM in lupin (*Lupinus polyphyllus*), pea (*Pisum sativum*), and lentil (*Lens culinaris*) increased the activity of cytochrome *c* oxidase in the cytosol, and decreased activity of this enzyme in seedling root mitochondria (*Ziółkowska et al., 2014*). The exact reason for these changes is not clear. There is no literature data concerning AOX functioning under DCF treatment, despite an increase in this enzyme's activity having been observed in plants' response to other stresses, such as high temperature, infection, or nutrient imbalance (*Simons et al., 1999*; *Escobar, Geisler & Rasmusson, 2006*; *Zalutskaya, Lapina & Ermilova, 2015*). Therefore, it was decided to examine both cytochrome *c* and AOX-pathway functioning in DCF-treated cells of the green alga *Chlamydomonas reinhardtii*. The scheme of potential influence ways of DCF treatment on the electron transport chain in *Chlamydomonas reinhardtii* cells has been illustrated on Fig. 1.

One of our earlier research projects showed that DCF affects cell cycle progression, delaying cell division as compared to non-treated cells (*Harshkova et al., 2021a*). Since the cell cycle progression seems to be tightly connected to changes in mitochondrial activity, number, shape, size, and cellular location (*Kianian & Kianian, 2014*), and the differences in respiration activity between young and mature cells may affect the sensitivity of cells to stress factors, it was decided to use synchronous *C. reinhardtii* cultures (*Pokora et al., 2017*, *2018*; *Majewska et al., 2018*; *Čížková et al., 2019*) to eliminate the influence of the cell developmental stage on the results.

Based on our and other researchers' experience (*Aksmann et al., 2014*; *Cross & Umen, 2015*; *Pokora et al., 2017*; *Čížková et al., 2019*; *Majewska et al., 2021*; *Harshkova et al., 2021a*, *2021b*), it was assumed that time points 0, 3, 6, and 9 h after the start of the light period of the cell cycle coincides with the following stages: zoospores, young cells, adult cells, and mother cells. Thus, oxygen consumption rate by the cells and mitochondrial activity parameters (mitochondrial membrane potential (MMP) and mitochondrial

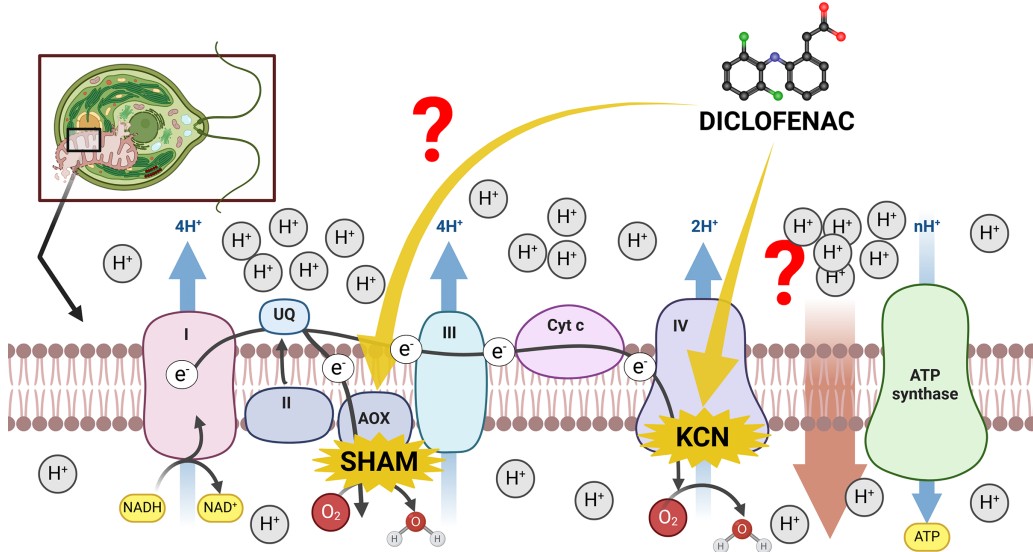

**Figure 1** **The scheme of electron transport chain in algal mitochondria with the sites of KCN and SHAM action marked.** I, II, III, IV = mitochondrial complexes I (NADH-ubiquinone oxidoreductase), II (succinate dehydrogenase), III (ubiquinol–cytochrome *c* oxidoreductase), and IV (cytochrome *c* oxidase); AOX, alternative oxidase; Cyt *c*, cytochrome *c*. Created with BioRender.com. Adapted from "Electron Transport Chain", by BioRender.com. Retrieved from https://app.biorender.com/biorender-templates. License: CC-BY-NC-ND.

reactive oxygen species (mtROS)) were analyzed at those time points. Additionally, for illustration of MMP change, confocal microscopy was used. Observation of the cell's structure organization by electron microscope was included in analyses to check for any malformation in mitochondrial structures during DCF exposure.

## MATERIALS AND METHODS

### Culture conditions and exposure to diclofenac

Wild-type *Chlamydomonas reinhardtii* strain CC-1690 (Chlamydomonas Resource Center, University of Minnesota, USA; https://www.chlamycollection.org/) was grown in 200-mL glass vessels, in liquid HSM (High Salt Medium; pH 6.9 ± 0.1) (*Harris, 2008*), at 30 °C. Cultures were aerated with sterilized air (PTFE filter, Sartorius 2000) enriched with 2.5% (v/v) $CO_2$ (*Harshkova et al., 2021b*). Photosynthetically active radiation intensity measured inside the culture vessels was 250 ± 5 μmol photons $m^{-2} \times s^{-1}$, provided by white, fluorescent tubes (OSRAM Dulux L, 55W040) (*Harshkova et al., 2021b*). Population growth was synchronized by alternating light and dark periods (L/D 10/14 h); before the beginning of each light period, the algal culture was diluted to a constant density of ca. $1.5 \times 10^6$ cells $\times mL^{-1}$ (*Pokora et al., 2018*). The cultures were observed with a light microscope (Leica DM 1000 LED, Germany; magnification 10 × 40) to monitor synchronization. The synchronized population of daughter cells was used as an inoculum for experimental cultures. At the beginning of each experiment, the algae were treated with diclofenac (DCF) (diclofenac sodium salt, sodium; 2-[2-(2,6-dichloroanilino)phenyl] acetate; 98% purity; ABRC, Niederzenz, Germany) dissolved in $ddH_2O$ and added to the

culture to a final concentration of 135.5 mg × L$^{-1}$, corresponding to the toxicological value EC50/24 (*Majewska et al., 2018*). The untreated population was set as a control. Cells were sampled at 0, 3, 6, or 9 h after the start of the light period of the cell cycle, which corresponded to the following phases of cell development: zoospores, young cells, adult cells, and mother cells, respectively (*Cross & Umen, 2015*).

For examination by electron and confocal microscope, the algae cells were cultured as described above, except for light/dark synchronization (*Harris, 2008*). Cultures with an initial cell density of ca. $1.5 \times 10^6$ cells × mL$^{-1}$ were divided into two sub-cultures: control or incubated with DCF in a final concentration of 135.5 mg × L$^{-1}$. The cells were sampled after 24 h incubation and prepared for electron and confocal microscopy as described below (see subchapters below).

## Population density and cell volume

The number and volume of cells were estimated using an electronic particle counter (Beckman Coulter Z2) run by dedicated software.

## Cell's oxygen consumption rate

Oxygen consumption rate was determined with a Clark-type oxygen electrode (Oxygraph, Hansatech Ltd., Taipei, UK). Before measurement, the cultures were darkened for 30 min to stop photosynthesis, and all further steps were performed under dim light or in darkness. One mL of cell suspension (about $1.5 \times 10^6$ cells × mL$^{-1}$) was sampled directly from the culture vessel and placed in a measuring chamber of the Oxygraph. The cell suspension was stirred continuously, and oxygen consumption measurements were carried out at 30 °C in total darkness. The oxygen consumption rate was expressed in nmol $O_2$ and recalculated per 1 million cells (nmol $O_2$ × $10^6$ cell$^{-1}$ × min$^{-1}$). The data were obtained from three independent experiments with at least two biological replicates of each type of sample.

## Mitochondrial membrane potential and mitochondrial ROS assessment

Measurement of mitochondrial membrane potential (MMP) *in vivo* was performed by JC-1-staining (5,5′,6,6′-tetrachloro-1,1′,3,3′-tetraethyl-imidacarbocyanine iodide, Sigma-Aldrich) based on protocol by *Harshkova, Zielińska & Aksmann (2019)*. The stock solution of the fluorochrome was prepared in DMSO (dimethyl sulfoxide, BioShop, Burlington, ON, Canada). The final JC-1 concentration in the incubation mixture was 3 μM. The final concentration of DMSO in the incubation mixture did not exceed 0.1% (v/v). At 0, 3, 6, and 9 h after the start of the light period of the cell cycle, samples of the cell suspension were collected. Before the sampling, the culture vessels were darkened for 30 min to stop photosynthetic processes. MMP was assessed according to *Harshkova, Zielińska & Aksmann (2019)*, with modification: cells were pelleted by centrifugation for 5 min at 460 g in black test tubes and resuspended in HSM (heated to 30 °C) to obtain $1 \times 10^6$ cells × mL$^{-1}$ for JC-1-staining. Measurement of MMP was performed after incubation for 20 min in black 98-well plates, using the spectrofluorometer Varioskan Flash Microplate Reader

(Thermo Fisher Scientific, Waltham, MA, USA). The excitation wavelength for JC-1 was 488 nm, and emission wavelengths were 538 and 596 nm (for monomers and oligomers of fluorochrome, respectively). MMP was represented as the oligomers/monomers fluorescence signal ratio and was expressed in an arbitrary unit (a.u.). The data were obtained from two independent experiments with two biological replications and with two technical replications of each sample measurement.

For confocal microscopy examination, the cell suspensions were centrifuged, and the pellet (about $1 \times 10^6$ cells $\times$ mL$^{-1}$) was resuspended in a small volume of HSM and incubated with 3 µM JC-1. Cells stained with JC-1 were fixed with 4% paraformaldehyde in HEPES buffer for 1 h at RT (*Craig & Avasthi, 2019*). Cells were visualized with Leica STELLARIS 5 WLL confocal microscopy with the Lightning module using FITC/TRITC ex/em. The photos presented are maximum projections taken from z-stacks with Lightning deconvolution combining signals from the green-fluorescent JC-1 monomers (absorption/emission maxima ~514/529 nm) and the red-fluorescent J-aggregates (emission maximum 590 nm). The mean fluorescence intensity of 10 representative cells of each experimental variant was quantified in triplicate using Leica Application Suite X. To visualize both J-monomers and J-aggregates combined with autofluorescence of chlorophyll in untreated and DCF-treated cells, 488 nm laser line excitation was used. Monomers were visualized at 536 nm, J-aggregates at 581 nm and chlorophyll at 676 nm emission wavelength for pinhole Airy calculation. Presented photos are maximum intensity projections of z-stacks taken with Lightning deconvolution module.

Mitochondrial ROS (mtROS) assessment was performed by MTO-staining (MitoTracker$^{TM}$ Orange CM-H2TMRos; Thermo Fisher Scientific, Waltham, MA, USA). Stock solutions of the fluorochrome were prepared in DMSO. The final fluorochrome concentration in the incubation mixture was 0.5 µM. The final concentration of DMSO in the incubation mixture did not exceed 0.1% (v/v). At 0, 3, 6, and 9 h after the start of the light period of the cell cycle, samples of the cell suspension were collected. Before the sampling, the culture vessels were darkened for 30 min to stop photosynthetic processes. Cells were pelleted by centrifugation for 5 min at 460 g in black test tubes and resuspended in HSM (heated to 30 °C) to obtain $5 \times 10^6$ cells $\times$ mL$^{-1}$. Measurement of mtROS was performed after incubation with MitoTracker$^{TM}$ Orange for 45 min in black 98-well plates, using the spectrofluorometer Varioskan Flash Microplate Reader (Thermo Fisher Scientific, Waltham, MA, USA). The excitation wavelength for MTO was 551 nm, and the emission wavelength was 576 nm. mtROS was expressed in an arbitrary unit (a.u.). The data were obtained from two independent experiments with two biological replications and with three technical replications of each sample measurement. For confocal microscopy examination of the mitochondria localization in *Chlamydomonas* cell, the cell suspensions were centrifuged, and the pellet (about $1 \times 10^6$ cells $\times$ mL$^{-1}$) was resuspended in a small volume of HSM with MTO-staining (ex/em: 551/576 nm setting was used). To combine stained mitochondria with chlorophyll autofluorescence, additionally 485 nm laser line was used, with 623 emission wavelength for pinhole Airy calculation. Presented photos are maximum intensity projections of z-stacks taken with the Lightning deconvolution module.

To inhibit cytochrome *c* oxidase activity, potassium cyanide (KCN; Avantor Performance Materials, Gliwice, Poland), dissolved in ddH$_2$O was used. To inhibit alternative oxidase (AOX) activity, salicylhydroxamic acid (SHAM; N,2-dihydroxybenzamide; Sigma-Aldrich, St. Louis, MO, USA) dissolved in DMSO was prepared. Cell suspensions (15 mL) taken from the culture vessel were placed in a black Falcon tube and incubated with 0.5 mM KCN (*Ghosh et al., 2016*) or with 3 mM SHAM (*Liu et al., 2021*) for 5 min at 30 °C. After the incubation, oxygen consumption rate, MMP and mtROS were measured as described above.

### Ultrastructure examination

For electron microscope examination, cell suspensions were centrifuged, and the pellet (about 8–10 × 10$^6$ cells × mL$^{-1}$) was fixed overnight with 2.5% glutaraldehyde (Polysciences) in 0.1 M sodium cacodylate buffer, pH 7.2, then post-fixed with 2% osmium tetroxide (Agar) in 0.1 M sodium cacodylate buffer. Further, cells were dehydrated with increasing concentrations of ethyl alcohol, infiltrated, and embedded in Epon 812 resin (Sigma-Aldrich, St. Louis, MO, USA). Ultra-sections (approximately 65 nm) were cut on Leica UC7. Sections were stained with uranyl acetate and lead citrate and examined with a Tecnai Spirit BioTWIN transmission electron microscope (FEI). The samples of cell suspensions were obtained from two independent experiments with at least two biological replications.

### Statistical analysis

Statistical analysis was performed using MS Excel 365 (Microsoft, Redmond, WA, USA) and Statistica 13.3 (StatSoft, Hamburg, Germany). All numerical data were given as means ± SD. Statistica 13.3 (StatSoft, Hamburg, Germany) was used to compute a correlation matrix to measure the strength of relationships between different variables and to compute the basic statistical and nonparametric Mann-Whitney U-test when the population could not be assumed to be normally distributed. A *p*-value < 0.05 was considered significant. A discriminant analysis was performed to integrally evaluate the metabolic effects of diclofenac and compare them to ETC inhibitors.

## RESULTS

### Population density and cell volume

The initial number of cells in the synchronic culture was set to 1.5 × 10$^6$ ± 0.44 × 10$^6$ cells × mL$^{-1}$. The number of cells remained unchanged for 9 h of the light period of the cell cycle in both the control and DCF-treated cultures (Table 1). The initial cell volume was 68.25 ± 8.06 fL, and in the control cultures reached about 540 ± 58.54 fL after 9 h. In DCF-treated cultures the final cell volume was 23% lower than in the control (Table 1). The observed decrease in the mother cells size suggests that DCF interferes with cells growth and development.

**Table 1 The cell volume and population density in control and DCF-treated *Chlamydomonas reinhardtii* cultures.**

|  | 0 h | 3 h | 6 h | 9 h |
|---|---|---|---|---|
| **Cell volume (fL)** | | | | |
| Control | 68.25 ± 8.06 | 149.90 ± 17.96 | 307.39 ± 45.66 | 539.61 ± 58.54 |
| DCF | 68.25 ± 8.06 | 135.15 ± 21.89 | 217.26 ± 63.29* | 414.85 ± 31.35* |
| **Number of cells (million cells per mL)** | | | | |
| Control | 1.54 ± 0.04 | 1.66 ± 0.14 | 1.80 ± 0.18 | 2.04 ± 0.25 |
| DCF | 1.54 ± 0.04 | 1.69 ± 0.10 | 1.77 ± 0.28 | 1.76 ± 0.24 |

**Note:**
DCF was applied to the culture at 0 h at a concentration of 135.5 mg × $L^{-1}$. Data are presented as mean ± SD. *Indicates statistically significant differences between control and treated populations ($p < 0.05$; Mann–Whitney U test; $n = 8$).

## Cell's oxygen consumption rate

To estimate oxygen consumption rate, both control and DCF-treated cells were sampled after 6 h of the cell cycle. In control cells, the oxygen consumption rate was 21.91 ± 7.75 nmol $O_2$ × $10^6$ $cell^{-1}$ × $min^{-1}$ (Fig. 2). The ETC-inhibitors, KCN and SHAM, caused a decrease in the oxygen consumption rate by 33–34% when applied separately and by 37% when applied in combination. In DCF-treated cells, the oxygen consumption rate reached about 150% of the control. Incubation of DCF-treated cells with KCN or SHAM caused a significant decrease in oxygen consumption rate, by 47% and 40%, respectively. The two inhibitors, when applied in combination, diminished oxygen consumption rate by 60% as compared to DCF-treated cells (Fig. 2). Since the increase in whole-cell oxygen consumption observed in DCF-treated cells is reduced by KCN and SHAM, it can be suggested that this increase is linked directly to mitochondrial activity.

## Mitochondrial membrane potential and mitochondrial ROS assessment

The value of MMP at the beginning of the experiment was 21.66 ± 2.52 a.u. After 3 h of cell growth, the MMP did not differ between control and DCF-treated cells, but after 6 and 9 h this parameter value was significantly lower in DCF-treated cells than in the control culture: 22.89 ± 11.54 a.u. in control cells and 10.33 ± 2.89 a.u. in DCF-treated cells after 6 h; 24.53 ± 7.96 a.u. in control cells and 8.17 ± 1.45 in DCF-treated cells after 9 h (Fig. 3 and Table S1). MMP values were further diminished by SHAM and KCN + SHAM after 9 h of the experiment, to 13.52 ± 3.30 a.u. and 15.55 ± 1.19 a.u. in control cells, respectively (Fig. 3 and Table S1).

To visualize the changes in membrane potential, cells treated with DCF for 24 h were stained with JC-1 and examined using confocal microscopy (Figs. 4 and S1). It was found that control cells exhibited strong red fluorescence ("C-red cells"), while in DCF-treated populations two fractions of cells could be seen, namely cells with a fluorescent signal similar to that of control cells ("DCF-red cells"), and cells exhibiting much weaker and more green fluorescence ("DCF-green cells") (Figs. 4 and S1). Measurements of fluorescence based on confocal microphotographs confirmed that "C-red cells" and "DCF-red cells" belong to the same statistical group while "DCF-green cells" significantly differ

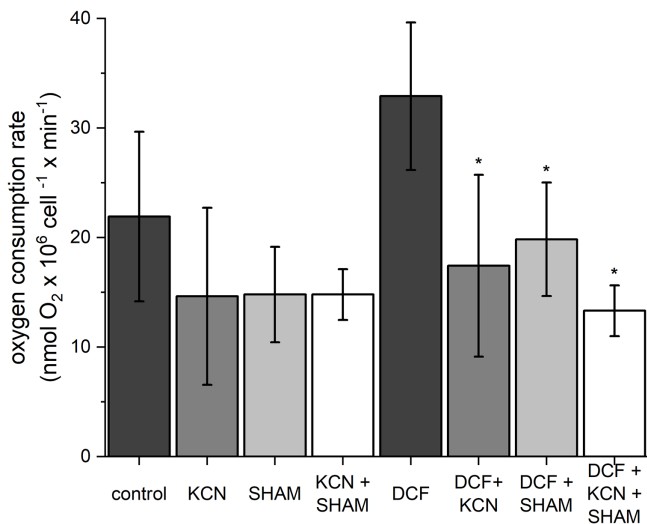

**Figure 2** **The influence of *ETC* inhibitors on oxygen consumption rate in control and DCF-treated cells of *Chlamydomonas reinhardtii* after 6 h from the beginning of light phase of the cell cycle.** DCF was applied to the culture at the beginning of the cell cycle (0 h) at a concentration of $135.5 \text{ mg} \times \text{L}^{-1}$. KCN, potassium cyanide-incubated cells; SHAM, salicylhydroxamic acid–incubated cells; DCF, cells treated with diclofenac. Data are given in $\text{nmol } O_2 \times 10^6 \text{ cell}^{-1} \times \text{min}^{-1} \pm$ SD. *Indicates statistically significant differences between DCF-treated cells and DCF-treated cells incubated with specific inhibitors, $p < 0.05$ (Mann–Whitney U test; $n = 8$).     

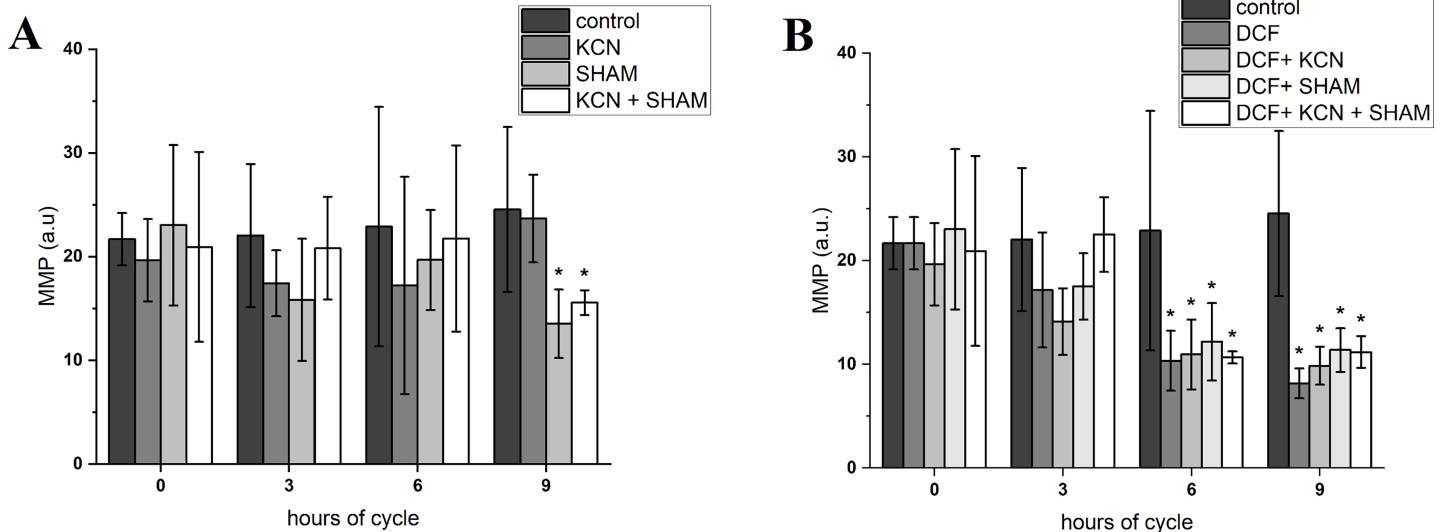

**Figure 3** **Mitochondrial membrane potential in control (A) and DCF-treated (B) *Chlamydomonas reinhardtii* cells.** DCF was applied to the culture at the beginning of the cell cycle (0 h) at a concentration of $135.5 \text{ mg} \times \text{L}^{-1}$. KCN, potassium cyanide-incubated cells; SHAM, salicylhy-droxamic acid–incubated cells; DCF, cells treated with diclofenac. Data are given in arbitrary units (a.u) of red/green fluorescence signal ratio (marked as MMP, a.u). Data are presented as mean ± SD. An asterisk (*) indicates statistically significant differences between control and treated populations, $p < 0.05$ (Mann–Whitney U test; $n = 8$).     

from them (results of fluorescence measurements are shown in Fig. S1). For a demonstration of the mitochondria localization in *Chlamydomonas* cell, confocal microphotographs were taken, showing JC-1 fluorescence combined with autofluorescence

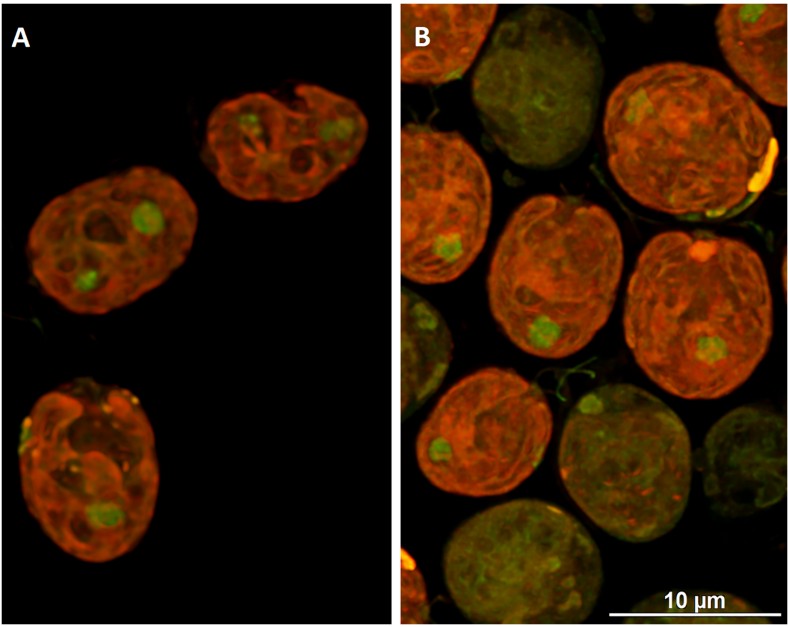

**Figure 4 Visualization of JC-1 fluorescence in *Chlamydomonas reinhardtii* cells, control (A) and treated with DCF (B).** DCF was applied to the culture at 0 h of the experiment at a concentration of 135.5 mg × L$^{-1}$. (A) Control population with cells exhibiting strong red-fluorescent J-aggregates ("C-red cells"; max. emission ~590 nm). (B) Cells treated with DCF for 24 h, with two different cell populations: "DCF-red cells", *i.e.*, cells with strong red fluorescence of J-aggregates (comparable to control), and "DCF-green cells" *i.e.*, cells with much weaker and mainly green fluorescence of J-monomers (max. emission ~529 nm). Photo credit: Małgorzata Kapusta.

of chlorophyll in untreated and DCF-treated cells (Fig. S2). Moreover, mitochondria was visualized using MTO (a positive control) (Fig. S3).

The initial (0 h) relative level of mtROS was 1.34 ± 0.33 a.u. After 3 h of the cell cycle a statistically significant decrease of mtROS in DCF-treated cells (0.95 ± 0.26 a.u.) was noticed, as compared to control cells (1.52 ± 0.22 a.u.). A similar trend was observed in cells sampled after 6 h of the experiment (1.74 ± 0.47 a.u. for control and 0.88 ± 0.09 a.u. for DCF-treated cells, respectively), and in cells sampled after 9 h of the cell cycle (1.13 ± 0.50 a.u. in control cells and 0.72 ± 0.15 a.u. in DCF-treated cells, respectively) (Table 2).

This data indicates that DCF treatment significantly disrupts mitochondrial function.

## Analysis of the cell's ultrastructure

Analysis of the cell's ultrastructure revealed that pyrenoid (Py), clearly visible in the control cells (Fig. 5A), was absent in DCF-treated cells, while the structure of thylakoids (th) seemed to be unchanged (Fig. 5B). In the chloroplasts of DCF-treated cells, higher numbers of large starch grains (s) could be seen (Fig. 5B). In DCF-treated cells vacuoles (v) were rare and seemed to be converted into larger autophagic vacuoles (av) (Fig. 5B).

Examination of the mitochondria (m) structure in more detail led to the conclusion that in the control cells, they were more regular, with correctly formed cristae (Fig. 6A). In DCF-treated cells, mitochondria were larger, irregular, and swollen with few and degraded cristae but with some inclusions or aggregates visible in the mitochondrial matrix

**Table 2 Relative level of mtROS in *Chlamydomonas reinhardtii* cells.**

|  | 0 h | 3 h | 6 h | 9 h |
|---|---|---|---|---|
| Control | $100.0 \pm 24.3$ | $100.0 \pm 14.2$ | $100.0 \pm 27.0$ | $100.0 \pm 43.9$ |
| DCF | $100.0 \pm 24.3$ | $62.5 \pm 27.3^*$ | $50.6 \pm 9.9^*$ | $64.0 \pm 20.3^*$ |

Note:
DCF was applied to the culture at 0 h of the experiment at a concentration of $135.5$ mg $\times$ L$^{-1}$. Data shown as % of control were originally expressed in a.u. of mtROS $\pm$ SD. $^*$Indicates statistically significant differences between control and treated populations ($p < 0.05$; Mann–Whitney U test; $n = 12$).

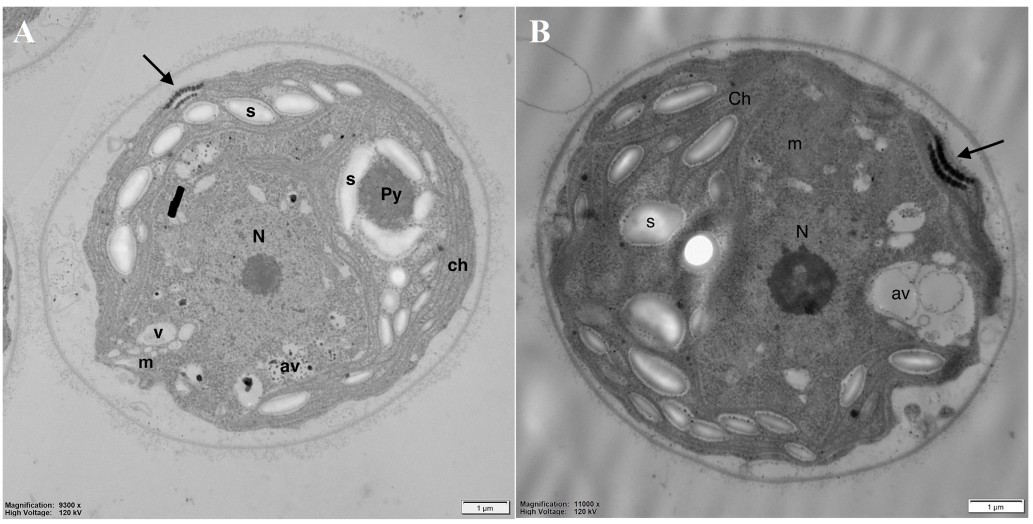

**Figure 5 Electron microphotographs of cell structures in control (A) and DCF-treated (B) *Chlamydomonas reinhardtii* cells.** DCF was applied to the culture at the beginning of the experiment (0 h) at a concentration of $135.5$ mg $\times$ L$^{-1}$ and cells were sampled after 24 h. Bar = 1 μm. N, nucleus; Ch, chloroplast; s, starch grain; Py, pyrenoid; m, mitochondria; v, vacuole; av, autophagic vacuole; black arrow, eyespot. Photo credit: Magdalena Narajczyk.

(Fig. 6B). These structural changes suggest that DCF treatment causes significant alterations in cellular organelles.

## Discriminant analysis and correlation matrices of selected cell parameters

Discriminant analysis and correlation matrices were created for three selected physiological parameters, characterizing cells collected after 6 h of the cell cycle: MMP, oxygen consumption rate, and cell volume. In the first step, the values of Wilks' lambda index were calculated for all parameters to select the parameter with the most important discriminatory contribution. The results indicated that MMP (Wilks' lambda = 0.3073) was the most influential. The statistical significance of the value ($p < 0.05$) indicates good discrimination between all experimental groups. Oxygen consumption made an additional discriminatory contribution (partial lambda index, $p < 0.05$) in all cultures (Table S3).

Further, all areas were checked for the variable's standardization coefficients to enable the construction of a reliable discriminant plot. MMP showed higher standardization coefficients for variables (0.7998) in the area "root 1 *vs.* root 2" compared to other areas

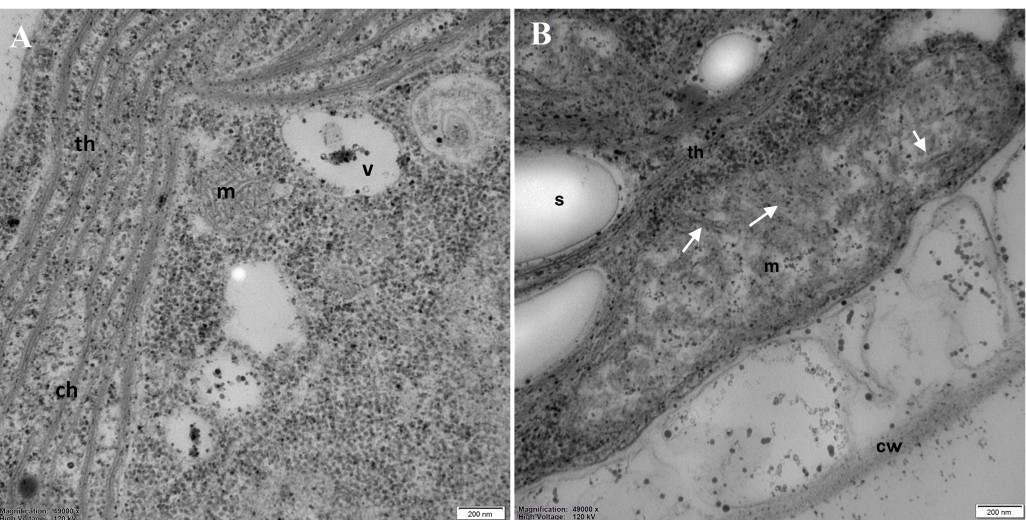

**Figure 6 Electron microphotographs of mitochondria structures in control (A) and DCF-treated (B)** *Chlamydomonas reinhardtii* **cells.** DCF was applied to the culture at the beginning of the cell cycle (0 h) at a concentration of 135.5 mg × L$^{-1}$, and cells were sampled after 24 h. Bar = 200 nm. Ch, chloroplast; s, starch grain; th, thylakoid; m, mitochondria; cw, cell wall; v, vacuole. Irregular structures of mitochondria with degraded cristae are marked as white arrows. Photo credit: Magdalena Narajczyk.

(Table S4). Figure 7 demonstrates the discriminant plot in the area "root 1 *vs*. root 2". It can be seen that the DCF-treated cells were evenly separated from all other groups (Fig. 7).

To determine the cell responses to DCF more precisely compared to control cells, after the discriminant analysis, each parameter was analyzed using correlation matrices (Spearman's test) separately. Comparing the DCF action to the action of ETC-inhibitors on the control cells did not reveal similarity (Table S2), except that for MMP Spearman's test showed a correlation (0.65, $p < 0.05$) between the cells incubated with both KCN and SHAM *vs*. DCF-treated cells without inhibitors (Table S5).

These statistical analyses did not reveal a similarity between cells' response to respiratory inhibitors and cells' response to DCF, which suggests that DCF impact on mitochondrial respiration is non-specific.

## DISCUSSION

The pollution of environment with NSAIDs is the emerging problem nowadays. The real environmental concentrations of pharmaceuticals are reported to be in the range of nanograms per liter, which is much lower than predicted environmental concentrations or predicted effective concentrations (ECs) for aquatic organisms (*Hejna, Kapuścińska & Aksmann, 2022*). Also, DCF investigated in the present work was shown to inhibit the growth of *Chlamydomonas reinhardtii* in a relatively high concentration with EC50 = 135.5 mg/L (*Majewska et al., 2018*), while its concentrations noted in water bodies reach up to 10,200 ng/L (*Hejna, Kapuścińska & Aksmann, 2022*). However, because of the continuous input of DCF from wastewater and its bioaccumulation potential, it can be

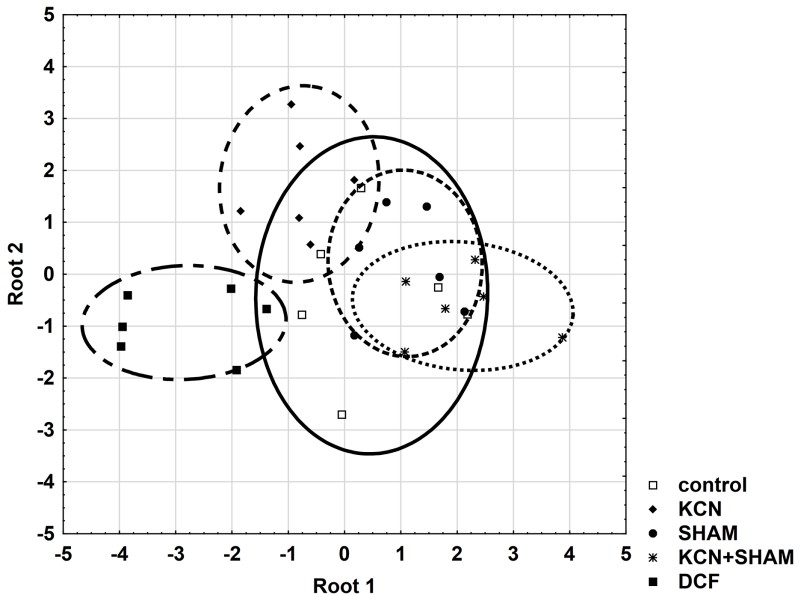

**Figure 7 Discriminant analysis of MMP, cell volume, and oxygen consumption in *Chlamydomonas reinhardtii* control cells and DCF-treated cells, incubated with *ETC*-blockers or non-blocked.** Cells were sampled at 6[th] h of the light phase of the cell cycle. KCN, potassium cyanide-treated cells; SHAM, salicylhydroxamic acid-treated cells; DCF, diclofenac-treated cells. The ellipses on the figure show the data distribution with a range of coefficient of 1.4.

assumed that the chronic effects of DCF on non-target organisms are stronger than acute effects. Moreover, drugs found in the environment usually occur as mixtures, which are known to be much more toxic than the individual substances (*Cleuvers, 2003*, *2004*). Thus, analysis of DCF toxicity, using EC values estimated in laboratory conditions, is an important component of ecotoxicological investigations.

Diclofenac has been reported as an inhibitor of population growth for microalgae species such as *Chlamydomonas reinhardtii*, *Desmodesmus subspicatus*, and *Dunaliella tertiolecta* (*Cleuvers, 2003*; *Lin, Yu & Lin, 2008*; *Majewska et al., 2018*). Some studies (*Hájková et al., 2019*; *Svobodníková et al., 2020*) indicated that photosynthesis disruption and oxidative stress induction in the cells were among the reasons for population growth inhibition by DCF. Similar observations were described in our earlier work (*Majewska et al., 2018*, *2021*). Moreover, it was reported (*Harshkova et al., 2021b*) that the treatment of *C. reinhardtii* with DCF in a concentration corresponding to the toxicological value EC50/24 can affect the growth of the single cell. A similar tendency was observed in the current work, namely the mature cells treated with DCF were smaller than the control ones. Since the decrease in the cell size suggests disruption of cellular energetics, analyses of mitochondria functioning seem to be a good direction for the investigations.

Interestingly, in the works by *Majewska et al. (2018)* and *Harshkova et al. (2021b)* no respiration inhibition was found in DCF-treated cells of *Chlamydomonas reinhardtii* when estimated based on the oxygen consumption rate, despite such effects having been reported for other organisms, including bacteria *Escherichia coli* (*Liwarska-Bizukojc, Galamon & Bernat, 2018*) and animal cells, *i.e.*, neuroblastoma cells (*Darendelioglu, 2020*) and

cardiomyocytes (*Ghosh et al., 2016*) during DCF-treatment. Unfortunately, there are few publications about dark respiration and mitochondrial functioning in microalgal cells treated with NSAIDs. One of the reasons could be difficulties in distinguishing between photosynthesis and respiration during investigations because of tight cooperation between chloroplasts and mitochondria, and the complexity of the energetics and oxygen metabolism in photoautotrophic cells.

In the present work, not only oxygen consumption rate was analyzed, but also the mitochondrial membrane potential was assessed using the fluorochrome JC-1, known for its sensitivity in analyzing mitochondrial efficiency (*Harshkova, Zielińska & Aksmann, 2019*). For revision of JC-1 staining suitability, the confocal microscope examination of *Chlamydomonas reinhardtii* cells was used (Figs. 4 and S2). Additionally, mitochondrial ROS levels were measured using the fluorochrome MitoTracker™ Orange CM-H2TMRos, a marker of mitochondrial oxidative stress (*Gonzalo et al., 2015*; *Martín-de-Lucía et al., 2018*). Considering that parameters of respiration efficiency (such as oxygen consumption rate, MMP, mitochondrial ultrastructure, *etc.*) in algal cells, as well as their sensitivity to stress factors, are strongly influenced by cell cycle phases (*Ehara, Osafune & Hase, 1995*), synchronous *C. reinhardtii* cultures were used for these investigations.

Analysis of the results indicated that even short-time exposure of the cells to DCF influences mitochondrial functioning. A tendency to increase in oxygen consumption rate suggested that DCF stimulates dark respiration (Fig. 2). One of the possible explanations for this observation is the disruption of mitochondria-chloroplasts cooperation. This assumption is supported by the literature data indicating that DCF negatively affects photosynthesis (*Majewska et al., 2018*, *2021*) and that mitochondrial activity is particularly important for maintaining life processes in cells with dysfunctional chloroplast (*Araújo, Nunes-Nesi & Fernie, 2014*, *Dang et al., 2014*; *Upadhyaya & Rao, 2019*). In *C. reinhardtii* cells, mitochondria-chloroplasts interdependence, involving redox control, energy balance, and organic compounds exchange, has been demonstrated in a wide range of works. To mention only a few examples, *Dang et al. (2014)* showed that in *C. reinhardtii* mutant *pgrl1*, deficient in PROTON GRADIENT REGULATION LIKE1 (PGRL1) protein, acting as ferredoxin-quinone reductase, mitochondrial cooperation and oxygen photoreduction downstream of PSI is crucial for maintaining biomass productivity. Further, it was shown that TOR kinase (target of rapamycin kinase) plays a key role in the regulation of both chloroplast and mitochondria functions (*Upadhyaya & Rao, 2019*). TOR kinase inhibition upon AZD-8055 treatment (selective TOR kinase inhibitor) leads to the dysregulation of chloroplast and mitochondria cooperation, an increase in respiration rate and mitochondria fragmentation, along with a decrease in photosynthetic activity (*Upadhyaya & Rao, 2019*). On the other hand, the tendency to increase in oxygen consumption rate observed in our work can result from mitochondrial membrane damage and the uncoupling effect, since increased oxygen uptake is strictly connected with mitochondrial uncoupling (*Barreto, Counago & Arruda, 2020*). Thus, to better explore this problem and to try to separate the processes occurring in the mitochondria from those occurring in the chloroplast, the MMP was analyzed.

It was found, that MMP tends to decrease after 3 h of exposure to DCF (Fig. 3 and Table S1) and this decrease under DCF treatment is compatible with reports which emphasized the loss of MMP in cells treated with another common NSAID, indomethacin (*Mazumder et al., 2019*). Visualization of JC-1-stained cells revealed, that in DCF-treated populations two fractions of cells could be seen, namely cells with a fluorescent signal similar to that of control cells, and cells exhibiting much weaker and more green fluorescence (Fig. S1). This suggests that the low MMP value noted on the population level (Fig. 3B) results from the appearance in the population of a fraction of cells with severely disturbed metabolism rather than from an equal reduction in the vital parameters of all cells in the population. It can be assumed that DCF strongly affects cells that experience mild metabolic or developmental disorders, not revealed under control conditions, but making these cells more susceptible to stress. Considering that analyzed algal populations are composed of millions of organisms, the assumption that some of them suffer from biochemical or physiological disorders seems plausible. This observation is supported by the investigations reported by *Harshkova et al. (2021a)*. In the cited work, analyses of *C. reinhardtii* cell cycle under DCF-induced stress strongly suggested, that some fraction of cells is eliminated from the population at the beginning of the experiment, and in surviving cells physiological functions are affected only slightly. Since the cited experiments were performed using synchronous cultures, the higher sensitivity of some organisms to DCF could not be linked to developmental stages of the cells. Thus, the oxygen consumption rate and MMP measurements described in the present work can be interpreted on population, not on a single cell level, which is important for ecotoxicological investigations where population functioning is the main goal of research.

The above statement is valid also for another important observation made in the present study, that the level of mtROS in DCF-treated populations was significantly lower than in control ones (Table 2). These observations were surprising in light of the literature reports that toxicity of many anthropogenic micropollutants, such as graphene oxide or poly (amidoamine) dendrimers, towards *C. reinhardtii*, is related to an increase in mitochondrial ROS formation accompanied by mitochondrial membrane depolarization and decreased mitochondrial activity (*Gonzalo et al., 2015*; *Martín-de-Lucía et al., 2018*).

Searching for a possible explanation for these observations, it was decided to take a closer look at the functioning of two ways of electron transport in the mitochondrial electron transport chain, because *C. reinhardtii*, like other plants, has a standard cytochrome *c* pathway of respiration as well as the alternative oxidase (AOX) pathway. The high activity of AOX results in energy dissipation as heat and MMP decrease, which on the one hand causes a reduction of ATP production (*Millenaar & Lambers, 2003*), but on the other hand plays an important role in stress response (*Zalutskaya, Lapina & Ermilova, 2015*). It was shown that AOX genes are upregulated under stress conditions such as $H_2O_2$, heat, high light illumination, nutrients limitation, and that AOX knockdown results in hypersensitivity to stress, as it was described for AOX-antisense lines of *Phaeodactylum tricornutum* (*Murik et al., 2019*). In the antisense line of this diatom photosynthesis was strongly affected, and AOX disfunction had a significant effect on gene expression and metabolome profile. Further, it was shown (*Mathy et al., 2010*) that

*C. reinhardtii* cells with reduced levels of *AOX1* (one of the AOX genes) exhibited hyper-reduction of the respiratory chain and elevated production of ROS compared to wild-type cells, as well as an increase activity of enzymes involved in anabolic pathways and a decrease activity of enzymes of the main catabolic pathways. Moreover, it was reported that in young cells the AOX-dependent electron transport is relatively low, while at advanced stages of cell development, AOX's relative contribution to oxygen consumption increases, so there may be differences in the sensitivity of young and mature cells to external toxic factors (for example DCF) depending on the leading respiratory pathways (*Strenkert et al., 2019*). Indeed, a comparison of the results of MMP measurements obtained from treatment with respiratory inhibitors allows us to conclude that mature cells are more sensitive to SHAM (AOX selective inhibitor) than the younger cells, and that this effect is further enhanced by DCF (Fig. 3 and Table S1). Because the response of respiration efficiency parameters (oxygen consumption rate and MMP) to selective respiratory inhibitors (KCN for cytochrome *c* oxidase and SHAM for AOX) and their response to DCF are not comparable (Figs. 2, 3 and Table S1), it can be suggested that DCF impact on mitochondrial respiration is non-specific. This suggestion is supported by statistical analyses, which did not reveal a similarity between cells' response to respiratory inhibitors applied separately, and cells' response to DCF (Fig. 7 and Table S5). The tendency for increased oxygen consumption rate (Fig. 2 in this study; (*Harshkova et al., 2021b*)) with low MMP and low mtROS production in DCF-treated populations may indicate an uncoupling of oxidative phosphorylation due to destruction of mitochondrial membranes. Destruction of mitochondrial membranes accompanied by uncoupling of oxidative phosphorylation has been documented for animal mitochondria treated with aspirin, indomethacin, naproxen, and piroxicam (*Somasundaram et al., 1997*). It cannot be also excluded that DCF causes mitochondrial swelling, resulting in the mitochondrial inner membrane permeability transition, as was reported for plant mitochondria subjected to anoxic stress (*Arpagaus, Rawyler & Braendle, 2002*; *Virolainen, Blokhina & Fagerstedt, 2002*). The phenomenon of mitochondrial swelling or hyper-fission (*Mazumder et al., 2019*) can explain our observation that in DCF-treated cells stained with JC-1 mitochondria has clearly distinguishable elements, while in control cells mitochondria are uniformly stained with the fluorochrom (Fig. S2).

To verify the abovementioned assumption about non-specific DCF impact on the mitochondria structure, the ultrastructure of *Chlamydomonas* cells was analyzed. To obtain a clear picture of the possible changes in the cell structure, a relatively long period of DCF treatment (24 h) was applied. The electron micrographs confirmed malformation in mitochondrial structures of cells suffering from DCF-induced stress (Figs. 5 and 6). The observations of elongated mitochondria, irregular with degraded cristae forms, support the suggestion that DCF causes mitochondrial swelling (*Arpagaus, Rawyler & Braendle, 2002*; *Virolainen, Blokhina & Fagerstedt, 2002*) or mitochondrial hyper-fission, as was reported for indomethacin-treated animal mitochondria (*Mazumder et al., 2019*). Ultrastructure analysis also revealed other DCF-induced changes, namely the disappearance of the pyrenoid in the chloroplast and the intensification of chloroplast starch deposition. The latter effect is consistent with the observation reported by

*Harshkova et al. (2021a)* that DCF causes a shift in material and energy balance toward carbohydrate storage (starch) in *C. reinhardtii* cells. The accumulation of starch in microalgal cultures under stress conditions also was reported by other authors (*Ivanov et al., 2021*). Thus, the mitochondrial dysfunctions observed in the present work may be one of the causes of previously reported cell developmental disorders and alterations in the cell cycle (*Harshkova et al., 2021a*).

## CONCLUSIONS

The results obtained in this work suggest that DCF strongly affects this fraction of cells, that have some developmental or metabolic dysfunctions, while more vital cells are affected only slightly, as it was already shown in the literature (see "Discussion"). In the cells suffering from DCF treatment, the drug influences on mitochondria functioning in a non-specific way, destroying the structure of mitochondrial membranes. This primary effect led to the uncoupling of oxidative phosphorylation. Since in algal cells mitochondria are important sources of metabolites, signaling molecules, and energy during both light and dark phases of the cell cycle, it can be assumed that mitochondrial dysfunction is an important factor in DCF phytotoxicity and impairment of cell development observed in other works (*Harshkova et al., 2021a*, *2021b*). Because studies of the effects of NSAIDs on the functioning of plant mitochondria are relatively scarce, the present work is an important contribution to the elucidation of the mechanism of NSAID toxicity toward non-target plant organisms.

### Funding

This work was supported by the National Science Centre, Poland [grant number UMO-2021/41/N/NZ8/00124]. The funders had no role in study design, data collection and analysis, decision to publish, or preparation of the manuscript.

### Grant Disclosures

The following grant information was disclosed by the authors:
National Science Centre, Poland: UMO-2021/41/N/NZ8/00124.

### Competing Interests

The authors declare that they have no competing interests.

### Author Contributions

- Darya Harshkova conceived and designed the experiments, performed the experiments, analyzed the data, prepared figures and/or tables, authored or reviewed drafts of the article, and approved the final draft.
- Elżbieta Zielińska performed the experiments, authored or reviewed drafts of the article, and approved the final draft.
- Magdalena Narajczyk performed the experiments, analyzed the data, prepared figures and/or tables, authored or reviewed drafts of the article, and approved the final draft.
- Małgorzata Kapusta performed the experiments, analyzed the data, prepared figures and/or tables, authored or reviewed drafts of the article, and approved the final draft.
- Anna Aksmann conceived and designed the experiments, analyzed the data, authored or reviewed drafts of the article, and approved the final draft.

## Data Availability

The raw data are available in the Supplemental File.

## Supplemental Information

Supplemental information for this article can be found online at http://dx.doi.org/10.7717/peerj.18005#supplemental-information.

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
