# Peer review of "Mitochondria dysfunction is one of the causes of diclofenac toxicity in the green alga Chlamydomonas reinhardtii"

_PeerJ, doi:10.7717/peerj.18005_

## Round 0.1 · original submission · Major Revisions

Improve clarity and precision of methods and results sections. Better explain, analyze, integrate, and discuss results. Add specific information to methods. Difficult to understand results, incomplete interpretation, and many results not discussed. Research is original but requires better analysis, integration, and interpretation. Please include a rebuttal letter in your revised submission.

Reviewer 1 ·

Basic reporting

-Title. The word “towards” is imprecise in the title context. An “in” may be used instead.
-The English edition. The authors must correct some grammatical, syntax, and spelling errors.
Some of them enlisted below:
Lane 94. Literature reports suggest that..
Lane 99. Has been observed…
Lane 155 and 228. Respiration is not measured in “intensity” units but in oxygen consumption rates. Please correct the whole manuscript.
Lane 170 and 184. ..assessed according to Harshkova, Zielinska & Aksmann (2019)
Lane 190. potassium cyanide (KCN; Avantor ….)
Lane 192. (SHAM; N,2-dihydroxybenzamide, Sigma-Aldrich) dissolved…
Lane 193. The DMSO description in methods should be included at the first mention and not in the last one, please correct.
Discussion subtitle is not in bold.

-Some parts of the manuscript include active voice, and others include passive voice. Passive voice is highly recommended throughout the manuscript; readers will focus on the action and the study results when using passive voice. A reader must not know who performed any action to develop this study. Please avoid “us”, “we”, or “our”.

Experimental design

Material and Methods. Additional information is needed to understand the methods and results better.
- The oxygen consumption rate should not be included as “mitochondrial activity determination” since it was measured in the whole cell; it should be reported as “cell´s oxygen consumption rate.”
The authors should explain or describe better:
- The measurement of mitochondrial membrane potential. The mitochondrial isolation protocol should be included in the manuscript; otherwise, the authors should clearly explain ¿how the fluorochrome JC-1 crosses the cell wall?, and ¿how the method specifically determines the MMP values?.
- How many replicates of each measurement and sample were performed to validate data statistically?
- Please use Standard deviation instead of std. errors in graphics.
- Lanes 171 and 185. What does HSM mean?
- Lane 227. Cell´s oxygen consumption rates
- Statistical differences in the results section text should be indicated as ( P< 0.05).

Validity of the findings

Results:
-Are the DFC concentrations as high as 135 mg/L in any environment? Concentrations of 10 micrograms/L of DFC affect growth, energy metabolism, redox balance, and the immune system in animals, mainly aquatic invertebrates (https://doi.org/10.1016/j.chemosphere.2021.133065).
-The authors should explain why these high concentrations of DFC were tested if they are not common in nature.
-The authors should explain how the KCN and SHAM inhibitors crossed the cell wall or entered the cells to specifically inhibit the mitochondrial complex IV and AOX and why they did not isolate mitochondria to perform their measurements in the specific organelle they wanted to analyze. The mitochondrial states 3 and 4 show different responses, and the respiratory control ratio may confirm the mitochondrial uncoupled state. The authors should mention the experimental state of their mitochondria (ADP presence or absence).
- The description of the results should be improved to be more precise.
- The legend in Figure 1 should be corrected. Respiration intensity is not a correct measure. The figure includes data about the oxygen consumption rates in C. reinhardtii cells. DCF-treated vs. control or in the presence of specific inhibitors of the mitochondrial complex IV and AOX. Please correct.
-Figure 2 needs to be significantly improved to be understood. Both graphs should be indicated as 2A and 2B for clarity. Data should not be presented as a percentage of the control but in the same units previously described in the text; otherwise, results are difficult to follow. The control bar, with means and SD, should be included at each time measurement in both figures since it is difficult to compare and interpret data.
- The authors should explain why the MMP values increase over control values after the additions of SHAM.
The legend of Figure 2 mentions: “(o) indicates statistically significant differences compared to the DCF-treated cells without blockers.” This is incorrect if comparing DCF + KCN, DCF+ SHAM, and DCF+ KCN + SHAM with the DCF control of each specific time and no with time 0h. Please explain or correct.
- A Figure or Table including the results of the relative level of mtROS; please include it.
- Since the result's explanation and interpretation need to be improved, discussions will vary after the analysis.

Reviewer 2 ·

Basic reporting

In this article, the authors examine the impacts of DFC on Chlamydomonas mitochondria. They conduct physiological analyses on Chlamydomonas cells at various growth stages, both with and without DFC. Their observations include changes in cell volume, disorganization of mitochondrial structure, and uncoupling of oxidative phosphorylation, indicating that mitochondrial dysfunction is an important factor in DCF phytotoxicity.
The introduction is clear. I only have a few minor comments. The experimental design has been carried out appropriately, and the investigation into DCF's effects on synchronized cells at different growth stages is interesting. My main concern is that the results section requires further refinement. There is ample opportunity for improvement in interpreting the results. More clarity and depth are needed, particularly in analyzing mitochondrial membrane potential, and additional explanation is required to address any observed surprising data.
In the discussion, comparing the authors' findings with existing data on Chlamydomonas mutants or stress conditions affecting mitochondria would be valuable. Incorporating this information into the discussion section could enhance it. For instance, I found it intriguing to juxtapose the data observed in this analysis with those documented during TOR kinase inhibition in Chlamydomonas (DOI: 10.1002/pld3.184).

Experimental design

The experimental plan was executed appropriately, but I would like to make a few remarks concerning certain points that could have been explored further.
While the introduction mentions limited information on the functioning of AOX under DCF treatment (line 98), raising our expectations for a more thorough investigation in this study, it is disappointing that no analysis has been conducted in this direction. For instance, examining AOX protein expression by immunoblotting at the analyzed time points would have provided informative insights into its relationship with ROS production.
Several fluorochromes were used to measure MMP or ROS. Incorporating confocal microscopic images with these fluorochromes to visualize the structure of mitochondria could have provided valuable information and would have strengthened the study.

Validity of the findings

In the first paragraph, the authors state a reduction in cell volume; it would have been desirable to show images of cells, control, and under DCF treatment to visualize this reduction in volume. Indeed, when examining both treated and untreated ultrastructure cells (Figure 3), there appears to be no apparent reduction in volume. This observation prompts questions about the reported (Table 1) reduction in cell volume. Also, additional commentary on the observed difference in volume would be interesting. It raises the question of whether treated cells exhibit a particular stage at which they become stagnant or "stuck."

It is surprising to note that for oxygen consumption and MMP measurements, there appears to be no further reduction in the presence of KCN+SHAM compared to KCN or SHAM alone. In some instances, there is even an increase. Could you please comment on this?

The authors assert that DCF-treated cells exhibit greater sensitivity to SHAM (line 336); however, Figures 1 and 2 suggest that they are more responsive to KCN up to a certain threshold. It would be beneficial to address this assertion in the Results section rather than solely in the Discussion to provide clarity. Presenting conclusions within the Results section can enhance comprehension and facilitate a more thorough understanding of the findings.

Unfortunately, Figure 5 may be challenging to interpret without visual cues such as colors or clear labels indicating which points correspond to MMP, cell volume, and oxygen consumption. Incorporating color-coded elements and clear legends would significantly improve the figure's clarity and help readers better understand the relationships between different variables.

The authors should include the raw data from the spectrofluorometer for MMP measurement and ROS assessment in the supplementary materials. This would enhance the study's transparency and reproducibility.

Additional comments

Introduction.
Line 71: it would be interesting to detail the changes in functioning in the mitochondria of animals.
Results:
Including diagrams would significantly aid readers unfamiliar with oxygen consumption and MMP measurement techniques. A straightforward diagram illustrating the electron transport chain (ETC) and the pathways inhibited by KCN and SHAM would be beneficial.
Figures 1 and 3: the DCF time treatment should be specified in the legend.
Table S2. should be in the main text.
Figure 4A. The contours of the mitochondrion are not clearly visible.

---

## Round 0.2 · Major Revisions

Please ensure that all review, editorial, and staff comments are addressed in a response letter and that any edits or clarifications are mentioned in the letter.

Reviewer 1 ·

Basic reporting

The manuscript aimed to determine the effect of diclofenac on the green algae C. reinhardtii. DCF toxicity was determined, and mitochondrial function was evaluated in different cell cycle stages, including the oxygen consumption rate, the mitochondrial membrane potential, and the ROS production. The data are valuable and interesting.

-Some language corrections were made, but the authors still need to check the English edition and correct grammatical, syntax, and spelling errors to improve the manuscript's clarity.
-SHAM and KCN are specific inhibitors to an enzyme, not blockers.
-The mitochondrial membrane potential is not reduced; it remains or is lost.
-The meaning of 6th h is not clear at all; it could be considered to be expressed only as after 6 h.
-Tables 3 to 5 should be considered as supplementary material.

Experimental design

-Material and Methods. Most comments were considered, and the missing information in the material and methods section was addressed.
-The results presentation should indicate in the corresponding figures and tables that means were graphed instead of medians since non-parametric tests were performed in some data sets.
Figure 2 shows large overlapping standard deviations that suggest no clear statistical differences that the authors describe in the manuscript. The figure legend does not make it easy to understand the statistical comparisons the circles indicate; please explain. The statistical analysis data should be included in the raw data file.
Figure 2. The Y-axis title needs to be correctly spelled.
Figure 3. The Y-axis units should be named in full in the title or the legend.

Validity of the findings

-Results in Figure 2 are difficult to interpret.
The authors state that DCF's impact on mitochondrial respiration is non-specific because both inhibitors did not affect the oxygen consumption rates of DCF-treated cells; however, the effect of both specific inhibitors (KCN, SHAM, and both) was not significant even in the control cells, whose COX and AOX are assumed to be active at this condition.
-The uncoupling state of mitochondria could only be confirmed by determining states 3 and 4 in the mitochondrial isolates by measuring the oxygen consumption rate; however, uncoupled mitochondria agree with the presented data. In addition, the DCF effect in the mitochondrial function should also be considered to induce mitochondrial swelling and the loss of cristae structure, which results in the mitochondrial inner membrane permeability transition; this may be considered to explain the significant change in the mitochondrial size and lost of structure.
- Results of the discriminant analysis are not discussed; please comment.

Additional comments

The manuscript is interesting and deserves improvement in some of the few aspects commented.

Reviewer 2 ·

Basic reporting

The discussion was improved, and answers were provided. Although the results section has been enhanced, each paragraph would benefit from a concluding sentence that highlights the relevant information. This concluding sentence would be a clear summary, aiding the reader's understanding.
For instance, in the first paragraph, it is mentioned that the DFC has an effect after 9 hours of treatment. A concluding sentence, such as 'The observed decrease in cell size suggests a disruption of cellular energetics', would have provided a clear summary and aided the reader's understanding.
In the second paragraph, a concluding sentence indicating that these results demonstrate an increase in whole-cell oxygen consumption and that SHAM and KCN treatments suggest this increase is linked to mitochondrial activity would be helpful. This approach should also be applied consistently across the other paragraphs.

Experimental design

It is great that you provided confocal microscopy images, but they raise several questions.
It would need additional explanation. Specifically, mitochondria should be indicated in the images. You mention that JC-1 is selective for mitochondria, yet the pictures show fluorescence throughout the entire cell (or possibly in the chloroplast?). Please clarify this. What does the green fluorescence correspond to? Consider adding an image with a mitotracker, as it could offer a helpful control.
I recommend revising this section prior to publication.

Validity of the findings

In lines 44–45 or 389–391, you state that "DFC strongly affects this fraction of cells that have some developmental or metabolic dysfunctions, while more vital cells are affected only slightly," based on the data obtained from confocal microscopy with the two populations observed. Are you suggesting that a fraction of the cells in your culture have developmental or metabolic dysfunctions? This is very surprising. Could this be instead linked to a specific developmental stage or cycle stage?
This conclusion should be moderate and instead discuss why these two populations are observed.

---

## Round 0.3 · accepted · Accept

Thank you for addressing the reviewers' suggestions. Your manuscript has been accepted by PeerJ.

Reviewer 1 ·

Basic reporting

ALL CONCERNS WERE CORRECTED

Experimental design

ALL CONCERNS WERE CORRECTED

Validity of the findings

ALL CONCERNS WERE CORRECTED

Additional comments

I HAVE REVISED THE NEW AND CORRECTED VERSION OF THE STUDY. ALL THE PREVIOUS SUGGESTED CHANGES WERE ADDRESSED IN THIS SECOND REVISION AND BETTER EXPLAINED THOSE THAT WERE UNCLEAR. THE RESULTS ARE INTERESTING AND VALUABLE AND NOW EASY TO UNDERSTAND. HENCE,  I CONSIDER THE MANUSCRIPT SHOULD BE ACCEPTED.